# Molecular Characterization, Expression Profiling, and SNP Analysis of the Porcine *RNF20* Gene

**DOI:** 10.3390/ani10050888

**Published:** 2020-05-20

**Authors:** Ying Zhao, Shulin Yang, Yanfang Wang, Cong Tao

**Affiliations:** 1State Key Laboratory of Animal Nutrition, Institute of Animal Sciences, Chinese Academy of Agricultural Sciences, Beijing 100193, China; zying_0317@126.com (Y.Z.); yangshulin@caas.cn (S.Y.); wangyanfang@caas.cn (Y.W.); 2College of Life Sciences, Qingdao Agricultural University, Qingdao 266109, China

**Keywords:** RNF20, SNP, porcine, PCR–RFLP, backfat thickness

## Abstract

**Simple Summary:**

In this study, we found that RNF20 is ubiquitously expressed in porcine tissues, and the sequence of the RING domain was highly conserved across different species. Eight potential single nucleotide polymorphisms (SNPs) were discovered, and one of them, SNP1 (A-1027G), was confirmed by PCR-restriction fragment length polymorphism (RFLP). Allele frequency differences were also analyzed in four pig breeds. This study provides a preliminary understanding of the porcine *RNF20* gene.

**Abstract:**

Fat deposition is considered an economically important trait in pig breeding programs. Ring finger protein 20 (RNF20), an E3 ubiquitin protein ligase, has been shown to be closely involved in adipogenesis in mice, suggesting its conserved role in pigs. In this study, we obtained the exon sequences of the porcine *RNF20* gene and characterized its molecular sequence. The porcine *RNF20* gene contains 20 exons that encode 975 amino acids, and its RING domain is highly conserved across different species. Western blot analysis revealed that RNF20 was widely expressed, especially in various fat depots, and the level of H2B monoubiquitination (H2Bub) was highly consistent. Eight potential SNPs were detected by sequencing pooled PCR fragments. PCR–RFLP was developed to detect a single nucleotide polymorphism (A-1027G) in exon 1, and the allele frequency differences were examined in four pig breeds. The G allele was predominant in these pigs. Association analysis between (A-1027G) and the backfat thickness of three commercial pig breeds was performed, but no significant association was found. Taken together, these results enabled us to undertake the molecular characterization, expression profiling, and SNP analysis of the porcine *RNF20* gene.

## 1. Introduction

Ring finger protein 20 (RNF20), an ortholog of yeast Brel 1a, is an E3 ligase that ubiquitinates histone H2B 120 lysine [1]. RNF20 and H2B monoubiquitination (H2Bub) have been shown to be important for a variety of biological processes, such as DNA replication [2], gene transcription [3,4], the DNA damage response [5,6], chromosome structure maintenance [7,8], the cell cycle [8,9,10], stem cell differentiation [10,11], tumorigenesis [12,13] and the inflammatory response [14]. Furthermore, it has been indicated that the RING domain of the RNF20 protein is key to modify histone H2BK120 [15,16]. Accumulating evidence has revealed the critical roles of RNF20 in lipid metabolism. *RNF20* overexpression repressed lipogenesis by inhibiting sterol regulatory element-binding protein 1c (SREBP1c), thereby suppressing hepatic lipid metabolism [13]. In particular, adenoviral overexpression of *RNF20* markedly reduced the level of triglycerides and decreased the lipogenic program in the liver [17]. A recent report found that Rnf20 is highly expressed in fat tissues from high-fat diet-fed mice compared to those from chow diet-fed mice, and *Rnf20* heterozygous mice (*Rnf20*^+/−^) exhibited significantly reduced fat mass compared to wild-type littermates [18]. Taken together, these data demonstrated that RNF20 plays a key role in fat deposition in rodents.

Fat deposition in pigs can be described, in general, with two representative phenotypic measures, backfat thickness (BFT) and intramuscular fat (IMF) percentage [19], and particularly, fat reduction is perceived as a decrease in BFT [20]. Pig breeding programs have a long history of focus on less fat deposition to meet people’s demands for lean meat. Therefore, identifying the genes and pathways that control this economic trait is important for developing molecular markers for selection. Over the past several decades, many locis that affect fat deposition and backfat thickness have been identified by association studies, linkage analysis, and GWAS [21]. However, fat metabolism in pigs is highly complex, and more related genes and pathways still need to be investigated. As described above, Rnf20 has been shown to be involved in adipogenesis in mice, suggesting its potential role in fat deposition in pigs.

In the present study, we obtained all exon sequences (Appendix A) of the porcine *RNF20* gene by PCR amplification and characterized its molecular properties via computational bioinformatic analysis. Furthermore, we examined the expression profile of this gene and identified SNP1 in exon 1. The allele frequency was detected in four pig breeds, including commercial breeds (Yorkshire, Duroc, and Landrace) and Min pigs, a native breed living in Northeast China with strong fat deposition ability. The preliminary association between SNP1 and BFT in commercial breeds was further examined.

## 2. Methods and Materials

### 2.1. Populations and DNA Samples

A total of 296 ear tissues were collected from four pig populations for genomic DNA extraction, including Yorkshire (*n* = 64), Landrace (*n* = 165), Duroc (*n* = 37), and Min Pig (*n* = 30). The Min pig is a native breed living in Northeast China. The unrelated pigs from three commercial populations were raised under the same standard conditions. The experimental animals and procedures were approved by the Experimental Animal Welfare and Ethical of Institute of Animal Sciences, Chinese Academy of Agricultural Sciences (No. IAS2017-4).

Genomic DNA was extracted by phenol-chloroform, precipitated with 25 μL of 2 M NaCl and two volumes of ethanol, and dissolved with 100 μL ddH_2_O. The quantity and purity of the genomic DNA samples were measured using a NanoDrop^TM^ 2000c spectrophotometer (Thermo Scientific, Waltham, MA, USA).

### 2.2. PCR Amplification

To obtain the predicted exon sequences and detect the putative single nucleotide polymorphisms (SNPs) in the porcine *RNF20* gene, 10 pooled DNA samples (2 μg per sample; 2 samples per breed, including Yorkshire, Landrace, Duroc, and Min, as we described above, and 2 DNA samples from Meishan pigs that we kept in our lab) were used as a PCR template. Ten pairs of primers were designed based on the putative pig *RNF20* sequence (NC_010443.5) by Primer 3 (Version 4) (http://bioinfo.ut.ee/primer3-0.4.0/). Amplicons of the primers covered all exons and partial introns of the *RNF20* gene (Table 1 and Figure 1). These primers were synthesized by Sangon Biotech (Shanghai, China). PCR amplification was performed as follows: a final volume of 20 μL containing 25 ng genomic DNA pool, 150 μM dNTP, 0.25 μM of each primer, and 1 U high fidelity Taq polymerase (TaKaRa, Tokyo, Japan) in the reaction buffer supplied by the manufacturer. The thermocycler profile was 95 °C for 5 min; 34 cycles of 95 °C for 30 s, 60 °C for 90 s, and 72 °C for 30 s, followed by a final extension step at 72 °C for 10 min, finally ending at 4 °C. The PCR products were purified and subjected to Sanger sequencing (sequencer model: 3730XL, Sangon Biotech, Shanghai, China) to identify potential mutation sites. SnapGene (GSL Biotech LLC, Chicago, IL, USA) was used for visualization of sequencing data.

### 2.3. Bioinformatic Analysis

A phylogenetic tree showing the genetic relationships between different animals was constructed by MEGA 7.0 software using a neighbor-joining (N-J) algorithm based on the RNF20 nucleotide sequences, including human (NM_019592.7), chimpanzee (XM_016961351.2), mouse (NM_001163263.1), rat (NM_001107929.2), porcine (XM_001926594.6), sheep (XM_004004043.3), goat (XM_005684254.3), rabbit (XM_008258916.2), cattle (NM_001081587.1), chicken (NM_001031434.1), and dog (XM_532018.5) from the NCBI database. The support for each node was calculated by a bootstrap test with 500 replicates. The RING domains of the RNF20 protein (921 to 966 residues) from different species were aligned using MEGA 7.0 software. Bioinformatic analysis was performed with the ExPASy online tool (https://web.Expasy.Org/protparam/) to calculate the molecular weight and the theoretical isoelectric point (pI).

### 2.4. Western Blot Analysis

Tissues used for Western blot are from a 6-month-old Meishan pig and kept in our lab. Total protein was extracted from spleen, lung, kidney, muscle, inguinal white adipose tissues (iWAT), perirenal white adipose tissues (pWAT), omental white adipose tissues (oWAT), and backfat tissues using T-PER™ Tissue Protein Extraction Reagent (Thermo Scientific, Waltham, MA, USA). Proteins were resolved with 10% Tris/glycine SDS-PAGE transferred to nitrocellulose membrane (Merck Millipore, Billerica, MA, USA) blocked with 5% fat-free milk for 2 h at temperature and incubated overnight at 4 °C with the following antibodies: RNF20 rabbit polyclonal antibody (catalog no. 21625) was purchased from Proteintech (Chicago, IL, USA), H2Bub antibody (catalog no. 5546) and β-tubulin antibody (catalog no. 2146) were purchased from Cell Signaling Technology (Danvers, MA, USA), and finally, horseradish peroxidase-conjugated secondary antibody (catalog no. 7074) was incubated for 40 min at room temperature. Western blot substrate was detected with a Tanon imaging system.

### 2.5. PCR–RFLP

The polymorphic BanI restriction site was found around SNP1(A-1027G), and the RFLP method was used for genotyping. In total, 296 DNA samples of unrelated animals from Yorkshire (*n* = 64), Landrace (*n* = 165), Duroc (*n* = 37), and Min pigs (*n* = 30) were genotyped. PCR products were digested in a total volume of 10 μL, containing 1 μL of 10 U Ban I restriction enzyme (R0118V, New England Biolabs, MA, USA), 1 μL 10× NE Buffer, and 8 μL of ddH_2_O at 37 °C for 60 min. Restriction fragments were examined by electrophoresis on 1% agarose gel with 1× TBE buffer.

### 2.6. Association Analysis

Association studies of different genotypes and BFT in three commercial pigs, including Yorkshire (*n* = 64), Landrace (*n* = 165), and Duroc (*n* = 37), were simply performed by GraphPad Prism 7 (GraphPad Software Inc, La Jolla, CA, USA). Statistical comparisons between two genotypes were made using the two-tailed Student’s *t*-test. Comparisons between three genotypes were made by one-way ANOVA followed by Tukey’s post hoc test. Backfat thickness is presented as the mean ± standard error mean (SEM).

## 3. Result

### 3.1. Exon Sequences of the Porcine RNF20 Gene

To obtain exon information of the porcine *RNF20* gene, we designed 10 pairs of primers (Table 1 and Figure 1) located in putative introns. The sequence of each PCR fragment was aligned to the predicted porcine *RNF20* sequences (XM_001926594.6) by MEGA 7.0, and exon information was then obtained. Sequence data revealed that porcine *RNF20* contained 2928-, 212-, and 899-bp coding sequences (CDSs) and 5′- and 3′- untranslated sequences (5′- and 3′-UTRs), respectively. The gene structure of *RNF20* is shown in Figure 1. The gene clearly contains 20 exons, and the translation initiation codon (ATG) is in exon 2. Since the primers were designed in introns, we found conserved GT/AG exon/intron junctions based on the sequencing data.

### 3.2. Bioinformatic Analysis of the RNF20 Gene

To further investigate the molecular characterization of the porcine *RNF20* gene, bioinformatic analysis was performed based on the CDS sequence. The CDS encodes 975 amino acid residues with a predicted molecular mass of 113.8 kDa and the pI of 5.74. The CDS shared 92.5% amino acid identity with human RNF20 (NM_019592.7) and 89.3% with mouse RNF20 (NM_001163263.1). Furthermore, a comparative phylogenetic tree was built with the RNF20 amino acid sequences across different species. As shown in Figure 2, the N-J phylogenetic tree was clustered into two groups: mammalian and poultry. This result indicated that porcine is most adjacent to sheep, goat, and cattle, according to the difference of genetic distance and junction point. It has been reported that RNF20 contains a RING domain, which is located at residues 922-961 at the C terminus. The alignments of the RING domain from 11 species are shown in Figure 3, suggesting that the sequences of the RING domain were highly conserved during evolution.

### 3.3. RNF20 Is Ubiquitously Expressed in Porcine Tissues

To explore the RNF20 expression pattern at the protein level in pigs, we examined its expression in various tissues from an adult Meishan pig by Western blot analysis. Our data revealed that RNF20 is widely expressed in peripheral tissues, including spleen, lung, kidney, and muscle (Figure 4). Specifically, we are interested in examining its expression in different fat depots, including inguinal white adipose tissues (iWAT), perirenal white adipose tissues (pWAT), omental white adipose tissues (oWAT), and backfat tissues. Our results showed that all WATs expressed RNF20, but the highest expression was found in oWAT (Figure 4). Meanwhile, the level of H2Bub was highly consistent with RNF20 (Figure 4).

### 3.4. Potential SNPs Identification

To identify the potential SNPs in the porcine *RNF20* gene, we investigated the sequencing data from ten PCR products. Eight “double peaks” were found in these PCR amplicons (Figure 5). An A to G mutation located -1027 bp upstream from the initiation codon of the porcine *RNF20* gene was named SNP1 (A-1027G). Shown in Figure 5, other potential SNPs were named SNP2 (C-975A), SNP3 (C-263T), SNP4 (G+146T), SNP5 (A+12711C), SNP6 (C+15355T), SNP7 (A+15383G), and SNP8 (T+15665A). We examined the locations of these potential SNPs and found that SNP1 (A-1027G) and SNP2 (C-975A) were in the 5′-UTR, whereas other SNPs were in various introns. As polymorphisms in the 5′-UTR might be involved in regulating gene transcription, we focused on SNP1(A-1027G) in the following studies.

### 3.5. Genotype and Allele Distribution of the RNF20 Gene

The detailed location of SNP1 (A-1027G) is shown in Figure 6a, and the BanI restriction site was found around it; thus, PCR-RFLP was developed to detect this polymorphism. Two distinct alleles were observed after BanI digestion: (1) allele A-fragment 1425 bp, (2) allele G-two fragments of 1240 and 185 bp (Figure 6b,c; 185 bp is too short to be detectable). Then, allele frequencies were detected in 296 unrelated animals, including Yorkshire (*n* = 64), Landrace (*n* = 165), Duroc (*n* = 37), and Min pigs (*n* = 30). The genotypes and allele frequency analysis are shown in Table 2. We found that the G allele was predominant in four pig breeds, the AA genotype was not found in either Landrace or Duroc, while Chinese Min pigs only exhibited the GG genotype.

### 3.6. Association Analysis of SNP1 with Backfat Thickness

To test whether SNP1 was related to backfat thickness (BFT), an association study was performed with three commercial pig breeds (Table 3). We found that the BFT in Yorkshire with genotype GG (13.87 ± 1.47 mm) was thicker than the BFT in AA (13.19 ± 0.51 mm) and AG (12.71 ± 0.57 mm) pigs. This trend was further found in Duroc, and the BFT of GG genotype pigs (12.73 ± 0.76 mm) was thicker than that of AG individuals (11.75 ± 0.92 mm). However, these differences did not reach significance.

## 4. Discussion

Accumulating evidence has shown that the *Rnf20* gene regulates fat metabolism in mice; however, its function in pigs has not been determined yet. As fat-related economic traits, such as BFT and lean meat percentage, are critical to determining pig production, it is of interest to clone the porcine *RNF20* gene and investigate its potential role in affecting fat-related traits.

In this study, we obtained all exon sequences of the porcine *RNF20* gene by sequence analysis of 10 PCR amplicons. Bioinformatic analysis based on predicted amino acid sequences revealed the high conservation of RNF20 across multiple species, which is consistent with previous reports [22]. Furthermore, we aligned the RING domain in the RNF20 protein and the highly conserved RING domain suggesting that its biological role should be conserved among different species. To date, several substrates catalyzed by RNF20 have been identified, including H2BK120 [1], NCoR1 [18], AP-2α [23], and SREBP1c [17], which are all strongly involved in fat metabolism. In addition, fat mass was significantly reduced in *Rnf20* knockout mice [18]. Therefore, we speculate that adipocyte RNF20 may also serve as a facilitator for fat deposition in porcine.

There are some differences in adipokines synthesis and secretion between visceral fat and subcutaneous adipose tissue. For example, visceral fat tissue shows higher concentrations of adiponectin, plasminogen activator inhibitor-1 (PAI-1), interleukin (IL-6), and angiotensinogen [24]. Abdominal visceral fat is strongly associated with metabolic abnormalities [25]. In this study, we found a higher expression of RNF20 in oWAT of pigs, indicating that adipocyte RNF20 might be important to maintain endocrine homeostasis. Consistent with our speculation, heterozygous knockout of *Rnf20* in mice leads to endocrine disorders [18].

Mammalian RNF20/RNF40 complexes function similarly to their yeast homology Bre1a/b as ubiquitin ligases in monoubiquitination of Lys-120 (the equivalent of Lys-123 in yeast) of histone H2B [26]. Cells lacking RNF20 or expressing H2B K120R, which lead to a lack of ubiquitin conjugation sites, showed a dramatically reduced level of ubiquitylation [16]. Moreover, overexpression and depletion of *RNF20* increased and decreased the global level of H2Bub, respectively, in humans [9,12,16,27] and mice [7,26,28]. In our study, Western blot analysis revealed that the level of H2Bub is consistent with RNF20. It suggested a key role for RNF20 and H2Bub in the regulation of porcine adipose tissue development. However, a causal relationship between endogenous porcine RNF20 and H2BK120 needs to be further determined.

We performed a preliminary association study between SNP1(A-1027G) and BFT in three commercial lines; however, the differences did not reach significance. The fact that the G allele was 100% in Min pigs, a Chinese native pig breed that had a thicker BFT than commercial pigs, together with the observation that Yorkshire and Duroc pigs with the GG genotype had a higher mean value of BFT than the AG genotype. In Duroc, it suggested that G allele is responsible for the thicker BFT. In Yorkshire, only four AA individuals were detected by RFLP; the relationship between AA genotypes and BFT needs to be confirmed in other large populations. In addition, it is of interest to evaluate the expression level of the *RNF20* gene, as well as its downstream adipogenesis-related genes, such as AP-2α, NCoR1, and SREBP1c, in the samples with the different genotypes, which may help to elucidate the effect of this G to A variation.

## 5. Conclusions

We obtained exon sequences of the porcine *RNF20* gene and found that it was highly conserved across different species. RNF20 is ubiquitously expressed in various porcine tissues, especially in different fat depots. SNP1 (A-1027G) was identified, and the G allele was predominant in the four pig breeds that we detected. The association analysis between SNP1 (A-1027G) and backfat in three commercial lines was performed, but the difference did not reach significance.

## Figures and Tables

**Figure 1 animals-10-00888-f001:**
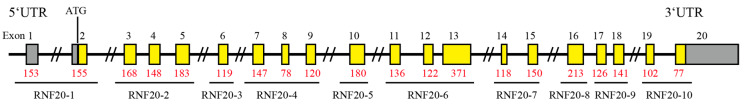
Structure of the porcine *RNF20* gene. Ten fragments used for PCR amplification are shown. This gene is organized into 20 exons (box) and 19 introns (horizontal lines). The left and right gray boxes represent the 5′- and 3′-UTRs, respectively.

**Figure 2 animals-10-00888-f002:**
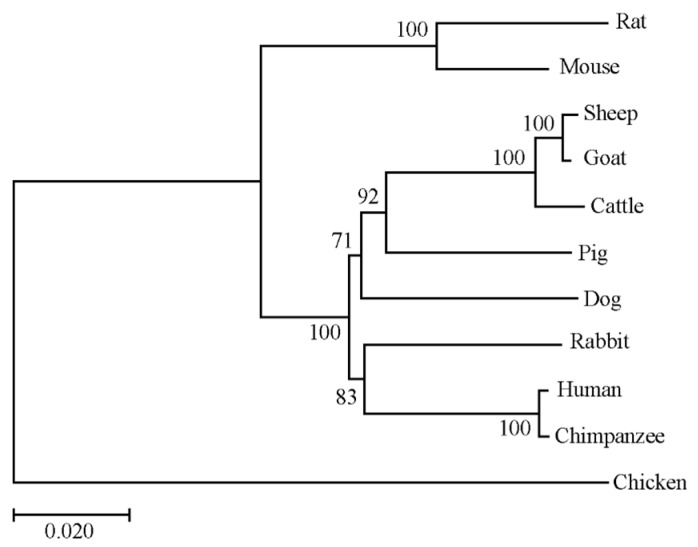
Phylogenetic tree of RNF20 protein sequences across different species. Animal species including human, chimpanzee, rat, mouse, sheep, goat, cattle, pig, dog, rabbit, and chicken. The horizontal branch lengths are proportional to the estimated divergence of the sequence from the branch point.

**Figure 3 animals-10-00888-f003:**
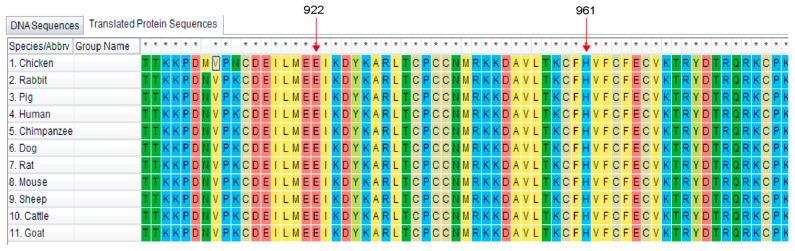
Alignment of RING domain sequences of different species. Note that the sequences of the RING domain (residues 922-961) are highly conserved across different species.

**Figure 4 animals-10-00888-f004:**
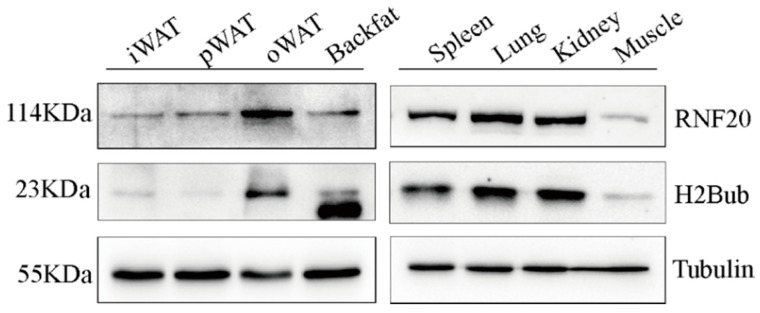
Expression profiles of RNF20 and the level of RNF20-mediated H2Bub in four fat depots (inguinal white adipose tissues (iWAT), perirenal white adipose tissues (pWAT), omental white adipose tissues (oWAT), and backfat) and four peripheral organs, including spleen, lung, kidney, and muscle from a 6-month-old Meishan pig. Tubulin was used as the loading control.

**Figure 5 animals-10-00888-f005:**
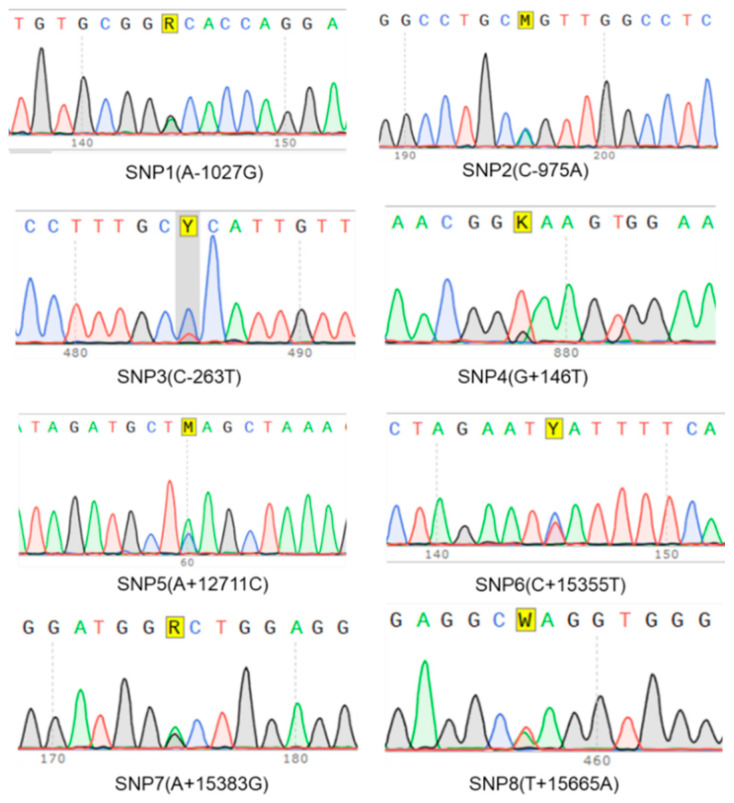
Eight potential single nucleotide polymorphisms (SNPs) were found by Sanger sequencing.

**Figure 6 animals-10-00888-f006:**
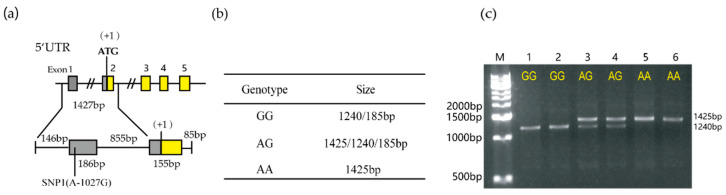
PCR-BanI-restriction fragment length polymorphism (RFLP) of the porcine *RNF20* gene. (**a**) SNP1 (A-1027G) was located 1027 bp upstream from the initiation codon of the *RNF20* gene in the 5′-UTR. (**b**) The three different PCR-RFLP genotypes of porcine *RNF20* gene. The sizes of GG, AA and AG are shown. (**c**) Analysis of digested PCR products with 1.5% agarose gel electrophoresis. The lane M:1 kb DNA ladder marker. Lanes 1 and 2: GG genotype; Lanes 3 and 4: AG genotype; Lanes 5 and 6: AA genotype.

**Table 1 animals-10-00888-t001:** Primers for the porcine *RNF20* gene.

Primers	Primer Sequences	Temp. (°C)	Product Sizes (bp)	Amplification Region
RNF20-1	F: TTTTCCTCTCCCTGACTCCTC	59.8	1427	Exon1/2
	R: TGTTACTCCAGAAGGCTTCCA			
RNF20-2	F: GTCAAGATTCCTCCCAGCTTC	60.5	1180	Exon3/4/5
	R: TGGCAGCTATAGTTCCGATCA			
RNF20-3	F: TAGCATGTCATTGCTTCGTTG	60.0	286	Exon6
	R: GACGAGCTTCAAAGCATTCAG			
RNF20-4	F: TGAACACCTCTCTTTGGGATG	59.1	960	Exon7/8/9
	R: TGTGTGTGAGCTAATCAGCAA			
RNF20-5	F: TAGCATCCAGGGCACAGATAC	60.1	389	Exon10
	R: GCCCACCATGTAGCAGAGTAA			
RNF20-6	F: TGTTGATTGGCTCCTTCTCTG	59.9	1275	Exon11/12/13
	R: GAACCAGACCACATGATAGCC			
RNF20-7	F: CTGATGCCCTTTGTTTTCTCA	59.4	1189	Exon14/15
	R: TCCTCTGATTCCAGAAAGCTC			
RNF20-8	F: TTGGGACTAGCAGATGCAGA	59.8	397	Exon16
	R: TTCCAAAATTCTGTTGAAGAG			
RNF20-9	F: AACAGATTTTTAGGCCTGTGG	60.0	591	Exon17/18
	R: TAGGTGGTTTCTGAACTGTGA			
RNF20-10	F: TGGGGAAGTGTGTAATGGGTA	60.2	600	Exon19/20
	R: TAGCCAGCTCGTCGTCTTCT			

**Table 2 animals-10-00888-t002:** Genotypic and allelic frequency distribution of SNP1(A-1027G) in four pig breeds.

Breeds	Number	Genotype Frequency (Number)			Allele Frequency	
		GG	AG	AA	G	A
Yorkshire	64	0.52 (33)	0.42 (27)	0.063 (4)	0.73	0.27
Landrace	165	0.88 (145)	0.12 (20)	0.00	0.94	0.06
Duroc	37	0.60 (22)	0.40 (15)	0.00	0.80	0.20
Min pig	30	1.00 (30)	0.00 (0)	0.00	1	0.00

**Table 3 animals-10-00888-t003:** Association analysis of porcine *RNF20* genotype with BFT in three pig breeds.

Breeds	Genotype (Number)	BFT (mm)	*p*-Value
Yorkshire	GG (33)	13.87 ± 1.47	GG-AG 0.7709
	AG (27)	12.71 ± 0.57	AG-AA 0.5337
	AA (4)	13.19 ± 0.51	AA-GG 0.6194
Landrace	GG (145)	12.89 ± 0.25	GG-AG 0.8668
	AG (20)	12.93 ± 0.69	
Duroc	GG (22)	12.73 ± 0.76	GG-AG 0.7606
	AG (15)	11.75 ± 0.92	

Values are presented as the mean ± SEM. BFT: backfat thickness.

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
