# Peer review of "Molecular Characterization, Expression Profiling, and SNP Analysis of the Porcine RNF20 Gene"

_animals, 2020, doi:10.3390/ani10050888_

Round 1

Reviewer 1 Report

After reading this manuscript and considering the possibility to start to make some corrections to the english form used, I made a simple analysis on the “juice” of what stated by Authors.

Apart from several unreadable periods, there are, according to me, three parts of the “juice” making the manuscript unacceptable.

  • A cDNA synthesis is certainly different from the PCR amplification of the exons of a gene identified by other researchers;
  • For the first time in my life I can see a chi square value for a population monomorphic for one allele. With the final consideration by Authors on the deviation from Hardy-Weinberg equilibrium because of the excess of GG homozygotes;
  • If differences between groups of your sample are not significant the consequence is that the two groups are not different and no dreaming considerations on “the correlation of the RNF20 gene with BFT can be done.

Author Response

1. A cDNA synthesis is certainly different from the PCR amplification of the exons of a gene identified by other researchers;

Response: thanks for this good point, we totally agree with this and we made the related correction of those descriptions throughout the manuscript (Line: 19; 124-125; 232)

2. For the first time in my life I can see a chi square value for a population monomorphic for one allele. With the final consideration by Authors on the deviation from Hardy-Weinberg equilibrium because of the excess of GG homozygotes;

Response: Actually, we only want to show the allele frequency and genotypes of pigs from different populations in the manuscript. Due to the low number of animals in each breed, chi square test is not necessary to be included here. We deleted the Chi value column in the revised version (Table 2).

3. If differences between groups of your sample are not significant the consequence is that the two groups are not different and no dreaming considerations on “the correlation of the RNF20 gene with BFT can be done.

Response: We agree with the reviewer and related descriptions have been revised.

Reviewer 2 Report

General Comments:
Congratulations for your work, it is very interesting and well written, easy to follow and understand all the procedure of the experiments, with the correct explanations of your decisions to proceed in this way. I really enjoyed the read.
I have just some minor revisions about some sentences and complete some parts on Materials and Methods and, discussion. I also included a suggestion for further studies even if will be great if you have the possibility to include that in the same work.

Author Response

General Comments:

Congratulations for your work, it is very interesting and well written, easy to follow and understand all the procedure of the experiments, with the correct explanations of your decisions to proceed in this way. I really enjoyed the read.

I have just some minor revisions about some sentences and complete some parts on Materials and Methods and, discussion. I also included a suggestion for further studies even if will be great if you have the possibility to include that in the same work.

Title:

1. I suggest a more specific Title focusing on the result for example: "Porcine RNF20 A-1027G variant gene is correlated with backfat thickness in pig".

Response: We appreciate this good suggestion. However, since this mutation was not significantly affect the backfat thickness significantly, might due to the low number of the population, we think it is not suitable to emphasize it in the title. We added the expression profile data in the revised manuscript and the revised title now is “Molecular Characterization, Expression Profile and SNP Analysis of Porcine RNF20 Gene”

2. Introductions:

Line 33: add acronym (RNF20), correct the comma with the point to finish the phrase.

Response: Accepted (Line 33).

3. Line42: copy-paste of "Particularly, aberrant expression of RNF20 leads to ectopic lipid accumulation in the liver [18]" from [19], please make it a bit different.

Response: we made the revision.

4. Line 43: High fat diet-fed "(HFD)" put it into backet and add the acronym for normal chow diet (NCD).

Response: Line43: We don't think it's necessary to add the acronym, because the words only appear once throughout the text.

5. Line61: I would eliminate this part of the sentences "was also performed", seem a repetition.

Response: Line61: Modified (Line 62)

6. Methods and Materials:

Line69: I would describe a bit the protocol followed for phenol-chloroform method, because the quantity of solution and tissue used are different in several papers.

Response: Based on this suggestion, we have added the information of extracting genomic DNA in line 71-72.

7. Line 99: please can you explain how did you prepare the pool for each breed and why you decide to include only 2 samples?

Response: we made the pool by this way: 2 μg DNA/pig (2 pigs/breed, including Yorkshire, Landrace, Duroc, Min and Meishan pigs), mix them together and use as a PCR template. Since we have five pig breeds (10 different DNA samples) included, variation should be big enough for identifying SNP loci (line 76-77).

8. Question: Did you deposit your sequences in Genbank and the SNPs found? There is no mention about it.

Response: We didn’t make the deposition yet, since here we just got the exon sequences by DNA PCR amplification, rather than cloned CDS from cDNA.

9. Results:

Line 120-121: Using the mixed pig DNA samples as template for PCR ("and" eliminate it) ten amplicons with the various lengths (1425, 1180, 286, 960, 389, 1275, 1189, 397, 591 and 600 bp, respectively) were obtained

Response: We made a revision for 3.1 based on reviewers’ suggestions (line 126-133). 

10. Line 178-179: In Duroc pigs, the frequencies of GG, AG were 0.595 and 0.405, respectively. no AA genotype? Please specify also in this case as for the Landrace.

Response: Yes, no AA genotype was found for Duroc, neither for Landrace. We made the description in the revised manuscript (line 192-193).

11. Discussion:

The fact that both Landrace and Duroc not presented AA genotype is an interested coincidence that, in my opinion deserve an explanation in this session describing the breed and meat characteristics versus Yorkshire and Min Pig. For me a brief description of the breeds analyzed in the introduction would be useful to make it more complete.

Suggestion further studies: would be interesting evaluate the expression level of the RNF20 gene by quantitative PCR, in the samples with the different genotype, in order to valuate your finding and make it stronger.

Response: We appreciate these valuable comments and suggestions. We added the breeds description in Introduction part (line 60-62), and association study description in Discussion part (line 231-232).

The suggestion for further studies is great and we will perform in the future.

Reviewer 3 Report

The manuscript focuses on porcine ring finger protein 20 (RNF20) and its potential association with backfat thickness (BFT). The topic is interesting – very little is known about this gene in pigs.

The language of the manuscript must be corrected. It requires also the English native speaker checking. There are some confusing, unclear sentences, wrongly used words and spelling mistakes.

The abstract is well constructed.

The introduction provides sufficient background for the study.
line 59: "we cloned porcine RNF20 cDNA" - there is no information in the manuscript about RNA isolation and reverse transcription. What the authors mean?

The methods are unclear and should be improved.
2.1. Populations and DNA samples
- lines 66-67: 64(YP) + 165(LP) + 37(DP) + 30(MP) = 296, not 300.
– Are the animals related or not? – the information about that is in the abstract and discussion and should be in the methods.
- Were the animals kept in the same piggeries?
2.2. Cloning CDS of porcine RNF20
- The section describes amplification and mentions sequencing. Did the authors cloned the DNA fragments or amplified? I guess that authors prepared amplicons for NGS sequencing. The word “cloning” is sometimes used with NGS, however in my opinion it is misleading.   
-How many PCR reactions were performed? What was exactly pooled? The 2.4 method section mentioned that the DNA was pooled - 2 samples per every breed. The Authors wrote that “approximately 150 ng of genomic DNA was used as template and (…) containing 25 ng genomic DNA pool, (…) 0.25 μM of each primer, and 1 U Taq polymerase (…)” There are four breeds. Is that mean that there were four DNA pools? It is not clear what exactly was prepared and sequenced.
- What sequencing method was used?  What equipment was used for sequencing?
2.3. Web-based bioinformatics analysis
- The authors should replace the NCBI Gene IDs with GeneBank Accession Numbers
- Did the Authors used MEGA 7.0 software to prepare alignment and to reveal coding sequences? It is not clear.

The results are presented in two tables and five figures.
All the figures (1-5) are really poor quality. They are hardly readable, full of visible pixels and they must be improved.
Did the Authors submitted their sequences in the GeneBank or other nucleotide data base / repository? I couldn’t find any accession number. The sequence/-s have to be submitted in GeneBank (or other nucleotide data base) ahead of the article publishing and the accession number should be presented.
3.1. Full-length coding sequencing (CDS) of porcine RNF20 gene
The Authors wrote that “sequence data revealed porcine RNF20 containing a 2,928 bp, 212bp, 899bp of coding sequences (CDS), 5’- and 3’- untranslated sequences (5’- and 3’-UTR), respectively”. Where are those sequencing data? The Authors should present such data (at least fragments) to support their findings, for instance in the Supplementary Materials.
4. The Authors wrote that “the full-length cDNA was then assembled based on pig RNF20 gene, which published in NCBI database in 2016” (lines 122-123). Why the authors didn’t compare their sequence and their RNF20 genomic structure with the porcine RNF20 model sequence and structure published in NCBI? How the Authors know that they have assembled the full-length cDNA sequence if they didn’t isolated RNA and didn’t sequenced the RNF20 transcripts?
3.2 Bioinformatics analysis of RNF20 gene
Figure 2 – some of the supports for nodes are too low (e.g. 23, 15, 31) – they do not support the branches; the basal part of the tree suggest polytomy. The phylogenetic tree prepared this way is misleading.   
3.3 Potential SNPs identification
The Authors found two SNPs in the 5’UTR and suggested that SNPs found in 5’UTR may have “potential role in regulating gene transcription”. But only one of the SNPs was selected -  SNP1 (A-1027G). Why? Why not both?
3.4 Genotype and allele distribution of RNF20 gene
There is no need to present the detailed frequency results in both table and text. The table is enough.
(lines 176-177): the same information is in the line 180.

In some parts the discussion presents the results without discussion. The discussion should be improved.

The references are well selected.

I am confused about the paper title. The most confusing were the words “cloning” and “cloned”, used several times in the manuscript and in the title. The method section 2.2, entitled “Cloning CDS of porcine RNF20” describes the DNA amplification, not the cloning. It is misleading. The title should be changed.
The results support the conclusion unsatisfactorily. The transcripts were not investigated. The phylogenetic tree is unnecessary or it should be improved. The authors investigated only one SNP although they found two SNPs in the 5’UTR. The other genes connected to backfat thickness were not included in the study. The studied population was small. In my opinion, after reading the paper, it is too soon to place a conclusion that this SNP deserves to be applied to MAS in pig breeding.

In conclusion, I think that the manuscript could be published in Animals only after major revision and English correction.

Author Response

The manuscript focuses on porcine ring finger protein 20 (RNF20) and its potential association with backfat thickness (BFT). The topic is interesting – very little is known about this gene in pigs.

The language of the manuscript must be corrected. It requires also the English native speaker checking. There are some confusing, unclear sentences, wrongly used words and spelling mistakes.

The abstract is well constructed.

The introduction provides sufficient background for the study.

1. line 59: "we cloned porcine RNF20 cDNA" - there is no information in the manuscript about RNA isolation and reverse transcription. What the authors mean?

Response: We have corrected the inaccurate description about cloning cDNA (Line: 19; 124-125; 232). We just obtained all exon sequences by DNA PCR amplicons.

2. The methods are unclear and should be improved.

- lines 66-67: 64(YP) + 165(LP) + 37(DP) + 30(MP) = 296, not 300.

Response: We have corrected the calculation errors (line 65-66).

3.  (1. Populations and DNA samples)

– Are the animals related or not? – the information about that is in the abstract and discussion and should be in the methods.

- Were the animals kept in the same piggeries?

Response: We have added “the unrelated pig populations were raised under the same standard conditions.” in the revised manuscript (line 67-68).

4.  (2. Cloning CDS of porcine RNF20)

- The section describes amplification and mentions sequencing. Did the authors cloned the DNA fragments or amplified? I guess that authors prepared amplicons for NGS sequencing. The word “cloning” is sometimes used with NGS, however in my opinion it is misleading.

Response: Yes, we agree, and the related description has been modified based on the comment (Line: 19; 124-125; 232).

5. -How many PCR reactions were performed? What was exactly pooled?

The 2.4 method section mentioned that the DNA was pooled - 2 samples per every breed. The Authors wrote that “approximately 150 ng of genomic DNA was used as template and (…) containing 25 ng genomic DNA pool, (…) 0.25 μM of each primer, and 1 U Taq polymerase (…)” There are four breeds. Is that mean that there were four DNA pools? It is not clear what exactly was prepared and sequenced.

Response: Thanks for this comment, we revised the manuscript as following “A pooled DNA samples (2 DNA samples/breed, including Yorkshire, Landrace, Duroc, Min and Meishan pigs; 2 μg/sample) were used as a PCR template to detect the putative SNPs. There is only one DNA pool (line 76-78).

6. - What sequencing method was used? What equipment was used for sequencing?

Response: PCR products were sent to Sangon Biotech (Shanghai, China) and analyzed by Sanger sequencing. The equipment model is 3730XL.

7. (3. Web-based bioinformatics analysis)

- The authors should replace the NCBI Gene IDs with GeneBank Accession Numbers

Response: Thanks for your advice, we have replaced Gene ID with GeneBank Accession Numbers in our revised manuscript.

8. - Did the Authors used MEGA 7.0 software to prepare alignment and to reveal coding sequences? It is not clear.

Response: Yes, we used MEGA 7.0 for RING domain alignment (line 93-94) and to reveal coding sequences (line 126-127).

9. The results are presented in two tables and five figures.

All the figures (1-5) are really poor quality. They are hardly readable, full of visible pixels and they must be improved.

Response: All pictures have been replaced with high quality.

10.  Did the Authors submit their sequences in the GeneBank or other nucleotide data base / repository? I couldn’t find any accession number. The sequence/-s have to be submitted in GeneBank (or other nucleotide data base) ahead of the article publishing and the accession number should be presented.

Response: We didn’t make the deposition yet since here we just got the exon sequences by DNA PCR amplification, rather than cloned CDS from cDNA. We provided the sequence as a supplementary Material. We think a cDNA synthesis is different from the PCR amplification of the exons of a gene. That is the reason we didn’t deposit them to GeneBank.

13. (3.1. Full-length coding sequencing (CDS) of porcine RNF20 gene)

The Authors wrote that “sequence data revealed porcine RNF20 containing a 2,928 bp, 212bp, 899bp of coding sequences (CDS), 5’- and 3’- untranslated sequences (5’- and 3’-UTR), respectively”. Where are those sequencing data? The Authors should present such data (at least fragments) to support their findings, for instance in the Supplementary Materials.

Response: Thanks, we listed the sequence as a Supplementary Material.

12.  The Authors wrote that “the full-length cDNA was then assembled based on pig RNF20 gene, which published in NCBI database in 2016” (lines 122-123). Why the authors didn’t compare their sequence and their RNF20 genomic structure with the porcine RNF20 model sequence and structure published in NCBI? How the Authors know that they have assembled the full-length cDNA sequence if they didn’t isolated RNA and didn’t sequenced the RNF20 transcripts?

Response: We revised the incorrect description in our revised manuscript, in fact, we did align our PCR sequences with predicted porcine RNF20 model sequence in NCBI, and got the gene structure as shown in Figure1. We revised “full-length cDNA sequence” to “exon sequences” throughout the manuscript (Line: 19; 124-125; 232).

13. (3.2 Bioinformatics analysis of RNF20 gene)

Figure 2 – some of the supports for nodes are too low (e.g. 23, 15, 31) – they do not support the branches; the basal part of the tree suggest polytomy. The phylogenetic tree prepared this way is misleading.

Response: We checked all the sequences for construction the tree carefully and rerun the essay, the revised N-J tree was shown in updated Figure 2.

14. 3.3 Potential SNPs identification

The Authors found two SNPs in the 5’UTR and suggested that SNPs found in 5’UTR may have “potential role in regulating gene transcription”. But only one of the SNPs was selected - SNP1 (A-1027G). Why? Why not both?

Response: Since we found the restriction enzyme site in SNP1, not in SNP2. For a preliminary essay, we just detected SNP1 with PCR-RFLP analysis.

15. 3.4 Genotype and allele distribution of RNF20 gene

There is no need to present the detailed frequency results in both table and text. The table is enough.

(lines 176-177): the same information is in the line 180.

Response: The redundant information was deleted.

16. In some parts the discussion presents the results without discussion. The discussion should be improved.

The references are well selected.

Response: we revised our discussion part and it is now not the repeat of results.

17. I am confused about the paper title. The most confusing were the words “cloning” and “cloned”, used several times in the manuscript and in the title. The method section 2.2, entitled “Cloning CDS of porcine RNF20” describes the DNA amplification, not the cloning. It is misleading. The title should be changed.

Response: The title was replaced in the revised version, “Molecular Characterization, Expression Profile and SNP Analysis of Porcine RNF20 Gene”.

18. The results support the conclusion unsatisfactorily. The transcripts were not investigated. The phylogenetic tree is unnecessary or it should be improved. The authors investigated only one SNP although they found two SNPs in the 5’UTR. The other genes connected to backfat thickness were not included in the study. The studied population was small. In my opinion, after reading the paper, it is too soon to place a conclusion that this SNP deserves to be applied to MAS in pig breeding.

In conclusion, I think that the manuscript could be published in Animals only after major revision and English correction.

Response: Thanks for all your comments. Based on these suggestions and comments, we revised the manuscript fully and written English was also improved. Some incorrect descriptions were removed or revised. Hopefully, all these modifications can satisfy the requirement of reviewer.

Reviewer 4 Report

In this study, Dr. Cong Tao and his colleagues cloned the porcine ring finger protein 20 (RNF20) gene. Furthermore, they study the polymorphisms of RNF20 and its correlation in phenotypes.  Based on their report, One single nucleotide polymorphism SNP1 (A-1027G) was discovered in exon 1. They found this potential correlation between SNP1 (A-1027G) and backfat thickness (BFT). In this article, they preassume that their finding can help persons improve and select the economic value in pigs. Its promising significance is to evaluate the characteristics. As a result, meat production and feeding efficiency will be improved. However, raised questions should be addressed. 

  1. This paper is in need of some editorial work respect to style and grammar.
  2. They must mention if their research is approved by animal welfare. I did not find a related description in the method and sampling. Am I correct?
  3. Due to this study related to polymorphisms of a gene, I do not know if they use high fidelity of polymerase (e.g, Phusion, Q5, or whatever high fidelity polymerase). However, I can not find related info in method. Pls confirm if they use high-performance enzyme. It is very important.
  4. They should use more ink in introducing the upstream regulators and downstream effectors of RNF20 gene. It will be helpful to show the significance of this research.
  5. I suggest that the author should deeply discuss the possible influence due to SNP1 (A-1027G).

What are the reasons probably causing the difference of the phenotype. They should introduce more signaling cascades involved in there. For example, signaling pathways about adipogenesis should be underscored, while it is also important to think about

Author Response

In this study, Dr. Cong Tao and his colleagues cloned the porcine ring finger protein 20 (RNF20) gene. Furthermore, they study the polymorphisms of RNF20 and its correlation in phenotypes.  Based on their report, One single nucleotide polymorphism SNP1 (A-1027G) was discovered in exon 1. They found this potential correlation between SNP1 (A-1027G) and backfat thickness (BFT). In this article, they preassume that their finding can help persons improve and select the economic value in pigs. Its promising significance is to evaluate the characteristics. As a result, meat production and feeding efficiency will be improved. However, raised questions should be addressed. 

  1. This paper is in need of some editorial work respect to style and grammar.

Response: we asked for help from an English native speaker to critically revise the manuscript.

  1. They must mention if their research is approved by animal welfare. I did not find a related description in the method and sampling. Am I correct?

Response: We have submitted an animal welfare certificate to editor. We have added the “Inspection form of Experimental Animal Welfare and Ethical of Institute of Animal Science, Chinese Academy of Agricultural Sciences (No. IAS2017-4)” in the revised manuscript (line 68-70).

  1. Due to this study related to polymorphisms of a gene, I do not know if they use high fidelity of polymerase (e.g, Phusion, Q5, or whatever high fidelity polymerase). However, I can not find related info in method. Pls confirm if they use high-performance enzyme. It is very important.

Response: Yes, we used high fidelity Taq polymerase for detection the polymorphisms, we added this information in Materials and Methods (line83)

  1. They should use more ink in introducing the upstream regulators and downstream effectors of RNF20 gene. It will be helpful to show the significance of this research.
  1. I suggest that the author should deeply discuss the possible influence due to SNP1 (A-1027G).

Response: we discussed the possible effect of this mutation (line 230-234).

  1. What are the reasons probably causing the difference of the phenotype. They should introduce more signaling cascades involved in there. For example, signaling pathways about adipogenesis should be underscored, while it is also important to think about

Response to 4-6: Based on your suggestion, we have made a major revision in the discussion section (229-236).

Round 2

Reviewer 1 Report

The paper is completely different from the first version. This is actually a brand new submission.

English language and style are worst than the first version.

The sum of the results is: a certain number of sequences, identification of polymorphic sites, Western blotting of different tissues of one single individual, no correlation with production traits. Full stop.

Western blot of individuals with different genotypes would have been certainly more interesting. At least for the two homozygotes.

Author Response

Comments and Suggestions for Authors

The paper is completely different from the first version. This is actually a brand new submission.

English language and style are worst than the first version.

The sum of the results is: a certain number of sequences, identification of polymorphic sites, Western blotting of different tissues of one single individual, no correlation with production traits. Full stop.

Western blot of individuals with different genotypes would have been certainly more interesting. At least for the two homozygotes.

Response:first of all, the written English was polished carefully by native English speaker and tracking changes can be found in revised manuscript. Thanks for the valuable suggestion, unfortunately, we don’t have the tissue samples of pigs with different genotypes at hand and we will perform it in the future. We discussed this in Line 251-254.

Reviewer 3 Report

The language of the manuscript still needs revision, especially in the newly (red) written fragments. I found grammar mistakes (e.g. wrongly used tenses). Did the English native speaker read the final manuscript version?

The abstract is well constructed. The introduction provides sufficient background for the study.

  1. line 59: "we cloned porcine RNF20 cDNA" - there is no information in the manuscript about RNA isolation and reverse transcription. What the authors mean?

Response: We have corrected the inaccurate description about cloning cDNA (Line: 19; 124-125; 232). We just obtained all exon sequences by DNA PCR amplicons.

OK

  1. The methods are unclear and should be improved.

- lines 66-67: 64(YP) + 165(LP) + 37(DP) + 30(MP) = 296, not 300.

Response: We have corrected the calculation errors (line 65-66).

OK

2.1. Populations and DNA samples)

– Are the animals related or not? – the information about that is in the abstract and discussion and should be in the methods.

- Were the animals kept in the same piggeries?

Response: We have added “the unrelated pig populations were raised under the same standard conditions.” in the revised manuscript (line 67-68).

OK, but now something is not clear with the breed count. How many breeds did the Authors finally use in the study 3, 4 or 5?

Line 25: “differences were examined in four pig breeds

Line 64:  “ear tissues were collected from four pig populations” – no information about Meishan

Lines 66-67: “The unrelated pigs from three commercial populations were raised under the same standard conditions

Lines 75-76: “pooled 10 DNA samples (2 μg per sample; 75 2 samples per breed, including Yorkshire, Landrace, Duroc, Min and Meishan pigs)” - 10/2=5

Please, standardize that or explain.

2.2.  (2. Cloning CDS of porcine RNF20)

- The section describes amplification and mentions sequencing. Did the authors cloned the DNA fragments or amplified? I guess that authors prepared amplicons for NGS sequencing. The word “cloning” is sometimes used with NGS, however in my opinion it is misleading.

Response: Yes, we agree, and the related description has been modified based on the comment (Line: 19; 124-125; 232).

OK

- How many PCR reactions were performed? What was exactly pooled?

The 2.4 method section mentioned that the DNA was pooled - 2 samples per every breed. The Authors wrote that “approximately 150 ng of genomic DNA was used as template and (…) containing 25 ng genomic DNA pool, (…) 0.25 μM of each primer, and 1 U Taq polymerase (…)” There are four breeds. Is that mean that there were four DNA pools? It is not clear what exactly was prepared and sequenced.

Response: Thanks for this comment, we revised the manuscript as following “A pooled DNA samples (2 DNA samples/breed, including Yorkshire, Landrace, Duroc, Min and Meishan pigs; 2 μg/sample) were used as a PCR template to detect the putative SNPs. There is only one DNA pool (line 76-78).

OK, but Meishan is not mentioned in section 2.1. Why?

- What sequencing method was used? What equipment was used for sequencing?

Response: PCR products were sent to Sangon Biotech (Shanghai, China) and analyzed by Sanger sequencing. The equipment model is 3730XL.

OK, please place the sequencer model in the Methods.

2.3. (3. Web-based bioinformatics analysis)

- The authors should replace the NCBI Gene IDs with GeneBank Accession Numbers

Response: Thanks for your advice, we have replaced Gene ID with GeneBank Accession Numbers in our revised manuscript.

OK.

However, you in line 91-92 the Authors mentioned “Horse (XM_014735936.2)”, but there is no horse in the figure 2. Did the Authors use equine sequence to prepare tree or not?

- Did the Authors used MEGA 7.0 software to prepare alignment and to reveal coding sequences? It is not clear.

Response: Yes, we used MEGA 7.0 for RING domain alignment (line 93-94) and to reveal coding sequences (line 126-127).

OK

3. The results are presented in two tables and five figures.

All the figures (1-5) are really poor quality. They are hardly readable, full of visible pixels and they must be improved.

Response: All pictures have been replaced with high quality.

Now the Figures look fine.

- Did the Authors submit their sequences in the GeneBank or other nucleotide data base / repository? I couldn’t find any accession number. The sequence/-s have to be submitted in GeneBank (or other nucleotide data base) ahead of the article publishing and the accession number should be presented.

Response: We didn’t make the deposition yet since here we just got the exon sequences by DNA PCR amplification, rather than cloned CDS from cDNA. We provided the sequence as a supplementary Material. We think a cDNA synthesis is different from the PCR amplification of the exons of a gene. That is the reason we didn’t deposit them to GeneBank.

OK, but in GeneBank you can also place genomic DNA sequences.

3.1. (3.1. Full-length coding sequencing (CDS) of porcine RNF20 gene)

- The Authors wrote that “sequence data revealed porcine RNF20 containing a 2,928 bp, 212bp, 899bp of coding sequences (CDS), 5’- and 3’- untranslated sequences (5’- and 3’-UTR), respectively”. Where are those sequencing data? The Authors should present such data (at least fragments) to support their findings, for instance in the Supplementary Materials.

Response: Thanks, we listed the sequence as a Supplementary Material.

OK

-  The Authors wrote that “the full-length cDNA was then assembled based on pig RNF20 gene, which published in NCBI database in 2016” (lines 122-123). Why the authors didn’t compare their sequence and their RNF20 genomic structure with the porcine RNF20 model sequence and structure published in NCBI? How the Authors know that they have assembled the full-length cDNA sequence if they didn’t isolated RNA and didn’t sequenced the RNF20 transcripts?

Response: We revised the incorrect description in our revised manuscript, in fact, we did align our PCR sequences with predicted porcine RNF20 model sequence in NCBI, and got the gene structure as shown in Figure1. We revised “full-length cDNA sequence” to “exon sequences” throughout the manuscript (Line: 19; 124-125; 232).

OK

3.2. (3.2 Bioinformatics analysis of RNF20 gene)

Figure 2 – some of the supports for nodes are too low (e.g. 23, 15, 31) – they do not support the branches; the basal part of the tree suggest polytomy. The phylogenetic tree prepared this way is misleading.

Response: We checked all the sequences for construction the tree carefully and rerun the essay, the revised N-J tree was shown in updated Figure 2.

OK

3.4. Potential SNPs identification

The Authors found two SNPs in the 5’UTR and suggested that SNPs found in 5’UTR may have “potential role in regulating gene transcription”. But only one of the SNPs was selected - SNP1 (A-1027G). Why? Why not both?

Response: Since we found the restriction enzyme site in SNP1, not in SNP2. For a preliminary essay, we just detected SNP1 with PCR-RFLP analysis.

OK

3.5 Genotype and allele distribution of RNF20 gene

There is no need to present the detailed frequency results in both table and text. The table is enough.

(lines 176-177): the same information is in the line 180.

Response: The redundant information was deleted.

OK

4. Discussion

In some parts the discussion presents the results without discussion. The discussion should be improved.

The references are well selected.

Response: we revised our discussion part and it is now not the repeat of results.

OK

General

- I am confused about the paper title. The most confusing were the words “cloning” and “cloned”, used several times in the manuscript and in the title. The method section 2.2, entitled “Cloning CDS of porcine RNF20” describes the DNA amplification, not the cloning. It is misleading. The title should be changed.

Response: The title was replaced in the revised version, “Molecular Characterization, Expression Profile and SNP Analysis of Porcine RNF20 Gene”.

The title is now acceptable.

- The results support the conclusion unsatisfactorily. The transcripts were not investigated. The phylogenetic tree is unnecessary or it should be improved. The authors investigated only one SNP although they found two SNPs in the 5’UTR. The other genes connected to backfat thickness were not included in the study. The studied population was small. In my opinion, after reading the paper, it is too soon to place a conclusion that this SNP deserves to be applied to MAS in pig breeding.

In conclusion, I think that the manuscript could be published in Animals only after major revision and English correction.

Response: Thanks for all your comments. Based on these suggestions and comments, we revised the manuscript fully and written English was also improved. Some incorrect descriptions were removed or revised. Hopefully, all these modifications can satisfy the requirement of reviewer.

Modification are fine. Now the results support the conclusions. In conclusion, I think that the manuscript can be published in Animals after minor revision and English correction.

Author Response

Comments and Suggestions for Authors

The language of the manuscript still needs revision, especially in the newly (red) written fragments. I found grammar mistakes (e.g. wrongly used tenses). Did the English native speaker read the final manuscript version?

Response: the written English was polished carefully by native English speaker this time and tracking changes can be found in revised manuscript.

The abstract is well constructed. The introduction provides sufficient background for the study.

  1. OK, but now something is not clear with the breed count. How many breeds did the Authors finally use in the study 3, 4 or 5?

Line 25: “differences were examined in four pig breeds

Line 64: “ear tissues were collected from four pig populations” – no information about Meishan

Lines 66-67: “The unrelated pigs from three commercial populations were raised under the same standard conditions

Lines 75-76: “pooled 10 DNA samples (2 μg per sample; 75 2 samples per breed, including Yorkshire, Landrace, Duroc, Min and Meishan pigs)” - 10/2=5

Please, standardize that or explain.

Response: we are sorry for such kind of confuse. We added the detailed sample information in each study in revised manuscript. We highlighted them in M&M part.

Line 84-87: this study aims to obtain the predicted exon and identify the potential SNPs, thus, more breeds included, the more potential variations. We used 10 individuals from 5 breeds, in which, DNA samples from Meishan pigs were kept in our lab.

Line 125-126: Genotyping was done with four pig breeds, including three commercial breeds and one Chinese native breed, the Min pig.

Line 131-132: Due to we don’t have the phenotype data of Min pigs, the association studies were only performed in three commercial populations, as shown in Table 3.

  1. OK, but Meishan is not mentioned in section 2.1. Why?

Response: We added this information in Line 85-87.

  1. OK, please place the sequencer model in the Methods.

Response: we added this information in Line 96.

  1. However, you in line 91-92 the Authors mentioned “Horse (XM_014735936.2)”, but there is no horse in the figure 2. Did the Authors use equine sequence to prepare tree or not?

Response: Sorry for the mistake, we have deleted this description in revised manuscript.

  1. OK, but in GeneBank you can also place genomic DNA sequences.

Response: Yes, we agree. As we include it as supplementary material and it doesn’t affect our conclusion. We are trying to clone the full-length of RNF20 genomic sequences and we will submit the entire sequence later.

Round 3

Reviewer 1 Report

The form was certainly improved and the paper is readable.

Author Response

The form was certainly improved and the paper is readable.

Response: Thanks for your positive comments on our manuscript.